# Antibiotic-induced population fluctuations and stochastic clearance of bacteria

**Jessica Coates[1†], Bo Ryoung Park[2†], Dai Le[2], Emrah Şimşek[2], Waqas Chaudhry[2], Minsu Kim[1,2,3]***

[1]Microbiology and Molecular Genetics Graduate Program, Graduate Division of Biological and Biomedical Sciences, Emory University, Atlanta, United States; [2]Department of Physics, Emory University, Atlanta, United States; [3]Emory Antibiotic Resistance Center, Emory University, Atlanta, United States

**Abstract** Effective antibiotic use that minimizes treatment failures remains a challenge. A better understanding of how bacterial populations respond to antibiotics is necessary. Previous studies of large bacterial populations established the deterministic framework of pharmacodynamics. Here, characterizing the dynamics of population extinction, we demonstrated the stochastic nature of eradicating bacteria with antibiotics. Antibiotics known to kill bacteria (bactericidal) induced population fluctuations. Thus, at high antibiotic concentrations, the dynamics of bacterial clearance were heterogeneous. At low concentrations, clearance still occurred with a non-zero probability. These striking outcomes of population fluctuations were well captured by our probabilistic model. Our model further suggested a strategy to facilitate eradication by increasing extinction probability. We experimentally tested this prediction for antibiotic-susceptible and clinically-isolated resistant bacteria. This new knowledge exposes fundamental limits in our ability to predict bacterial eradication. Additionally, it demonstrates the potential of using antibiotic concentrations that were previously deemed inefficacious to eradicate bacteria.
DOI: https://doi.org/10.7554/eLife.32976.001

**\*For correspondence:**
minsu.kim@emory.edu

[†]These authors contributed equally to this work

**Competing interests:** The authors declare that no competing interests exist.

## Introduction

The frequent failure of antibiotic treatments is a serious public health threat. A recent study projects treatment failures caused by antibiotic resistance will lead to 300 million deaths and a healthcare burden of $100 trillion by 2050 (*O'Neill, 2016*). This epidemic is further exacerbated by our inability to reliably eradicate antibiotic-susceptible bacteria. For example, antibiotic treatments of infections caused by antibiotic-susceptible bacteria never achieve a success rate of 100%, often failing to eradicate them unexpectedly (*Doern and Brecher, 2011*; *Weidner et al., 1999*; *Gopal et al., 1976*; *Ficnar et al., 1997*; *Forrest et al., 1993*). To design effective treatments and avoid antibiotic failure, there is a strong need to better understand the dynamics of bacterial populations exposed to antibiotics.

Previously, laboratory studies have extensively characterized how large bacterial populations (e.g., $\sim 10^8$ cells in a culture) decline under antibiotic treatment; e.g., see (*Nielsen et al., 2011*; *Ferro et al., 2015*; *Regoes et al., 2004*). These studies have led to the current, deterministic model of the pharmacodynamics, that is, the population dynamics of bacteria exposed to antibiotics follows a predetermined course and can be predicted deterministically *a priori*; see (*Regoes et al., 2004*; *Czock et al., 2009*) and references therein. This deterministic framework successfully captures the reproducible dynamics of a large bacterial population declining to a small population under antibiotic treatments. However, due to their experimental detection limits (e.g., >>100 cells [*Nielsen et al., 2011*; *Ferro et al., 2015*]), the dynamics of a small population undergoing extinction have not been directly characterized.

Inoculum size as small as a few cells can produce infections (*Jones et al., 2006*; *Haas and Rose, 1994*; *Jones et al., 2005*; *Tuttle et al., 1999*; *DuPont et al., 1989*; *Hara-Kudo and Takatori, 2011*; *Kaiser et al., 1992*). Thus, if antibiotics manage to reduce a large bacterial population to a very small population but fail to eradicate it, the survivors may replicate and restore infections. Additionally, these survivors are more likely to develop antibiotic resistance, making subsequent antibiotic treatment of the restored population more difficult (*Gullberg et al., 2011*; *Kohanski et al., 2010*; *Lopatkin et al., 2016*; *Dagan et al., 2001*; *Allen et al., 2014*). Thus, in many cases, including life-threatening infections or even minor infections in immuno-compromised patients, treatment success depends on complete clearance of the infection-causing bacteria. To effectively clear bacteria using antibiotics, it is critical that we understand not only how a large population of bacteria declines to a small population, but also how a small population eventually goes extinct. Extensive studies focused on the former process (discussed above). The present study focuses on the latter process.

By employing a conventional plate assay, single-cell microscopy, and quantitative modeling, we directly characterized the extinction dynamics of antibiotic-susceptible *Escherichia coli* populations. We found that antibiotics known to kill bacteria (i.e., bactericidal drugs) induce population fluctuations. At high drug concentrations, all populations go extinct (as expected), but the extinction time is highly variable and cannot be deterministically predicted *a priori*. Even at low drug concentrations, due to these fluctuations, populations go extinct with a non-zero probability. We found that the Markovian birth-death model quantitatively accounted for the probabilistic occurrence of population extinction. Informed by the model, we then altered the extinction probability by manipulating cell growth and showed that a bacterial population could be eradicated at low drug concentrations that were previously deemed inefficacious. Our work demonstrates that the deterministic knowledge obtained from previous studies of large bacterial populations cannot be extrapolated to population extinction. Our findings also have significant implications for the prediction of treatment outcomes, development of innovative therapies, and assessment of antibiotic efficacy.

## Results

### Contrasting trends in plating efficiency for bacteriostatic and bactericidal drugs

Previous studies of large populations have established the 'minimum inhibitory concentration' (MIC; the lowest concentration of the drug that inhibits population growth) as the most critical parameter for characterizing the dynamics of a bacterial population under antibiotics (*Regoes et al., 2004*; *Czock et al., 2009*; *Craig, 1998*; *Falagas et al., 2012*). The dynamics of bacterial populations exposed to different concentrations of antibiotics have been examined and modeled deterministically in relation to the MIC, as follows. First, without drugs, the growth rate of cells, $\lambda$, is higher than the death rate, $\phi$ (i.e., $\lambda > \phi$), and thus a bacterial population always grows. When drug concentration increases, as long as the concentration remains below the MIC (i.e., sub-MIC), growth rate is higher than death rate ($\lambda > \phi$), and thus a population still grows, albeit at slower rates. When the drug concentration increases further and reaches the MIC, growth rate becomes equal to death rate ($\lambda = \phi$), and the population size is maintained at a constant level. Only at drug concentrations above the MIC does a bacterial population decline. Extrapolating this deterministic knowledge to population extinction, studies often claimed that maintaining drug concentrations above the MIC was absolutely essential to eradicate bacterial populations. As will become evident later, our data challenge this deterministic framework.

As a first step to examine the dynamics of a small population in relation to the MIC, we used a plate assay and characterized how single *E. coli* cells grew and formed colonies at various antibiotic concentrations. Antibiotic-susceptible, wild-type (WT) *E. coli* cells were cultured in liquid LB medium without antibiotics and then spread on LB agar plates containing increasing concentrations of antibiotics. After 18 hr of incubation, the number of colony-forming units ($N_{CFU}$) was determined. By normalizing the $N_{CFU}$ to that for an antibiotic-free plate ($N^0_{CFU}$), we then obtained the plating efficiency ($=N_{CFU}/N^0_{CFU}$), which indicates the fraction of cells forming colonies. Following the definition of MIC as the lowest drug concentration that inhibits population growth, the lowest concentration yielding no visible colonies on the plates was defined as the MIC here. See *Figure 1—figure supplement 1*

for a detailed illustration of this procedure and *Supplementary file 1* for the MIC values for all of the antibiotics examined.

When we performed this plate assay for various antibiotic drugs, we observed two strikingly distinct trends, which depended on whether the drug used was bacteriostatic (which suppresses cell growth) or bactericidal (which induces cell death). For bacteriostatic drugs, at increasing concentrations, the plating efficiency remained nearly constant and abruptly dropped to zero when the drug concentration reached the MIC (*Figure 1A*); the grey line was obtained from a linear regression analysis of the whole data set below $0.75 \times$ MIC (see *Figure 1A* caption for details). This trend indicates that almost every single cell spread on the plate grew and formed colonies at a wide range of sub-MIC drug concentration, and no cells formed colonies at (and above) MIC. This observation, suggesting homogeneous population dynamics, agrees with the deterministic prediction discussed above. Additionally, we observed a decrease in colony size at increasing drug concentrations (*Figure 1—figure supplement 2*).

For bactericidal drugs, at increasing concentrations, the plating efficiency decreased *gradually* from 1 to 0 (*Figure 1B*); the grey line was obtained from a linear regression analysis of the whole data set (see *Figure 1B* caption for details). This trend contrasts with our finding for bacteriostatic drugs (compare the grey lines in *Figure 1A and B*) and cannot be explained by the deterministic model. In the literature, other studies have reported a similar gradual decrease in the plating efficiency (*Liu et al., 2011*; *Ernst et al., 2002*; *Dong et al., 2000*). However, those studies primarily concerned how to better determine the MIC in the face of such a gradual decrease, and have not characterized population dynamics underlying the gradual decrease.

## A subsequent plate assay reveals a lack of heritable resistance

The plating efficiency between 0 and 1 indicates heterogeneous colony formation. In the plate assay above, we found that at ~$0.6 \times$ MIC, the plating efficiency was ~0.5, meaning that approximately 50% of the cells plated formed colonies and 50% did not. One possible explanation is that the colony-forming cells were intrinsically more resistant to the drugs than the cells that did not form colonies, subsequently giving rise to resistant daughter cells (i.e., heritable resistance). To examine this possibility, for each bactericidal drug used in the experiment (for which the results are shown in *Figure 1B*), we picked colonies from agar plates exhibiting a plating efficiency of ~0.5 (near $0.6 \times$ MIC), suspended them in liquid medium, and immediately plated them on fresh agar plates containing various concentrations of the same drug. The results were plotted in *Figure 1C* and *Figure 1—figure supplement 3*. Contrary to our expectation, the plating efficiency of the second plating was about the same as that of the first plating, or in some cases, marginally lower (possibly because cells were challenged with drugs twice consecutively). This observation rules out heritable resistance as an explanation for heterogeneous colony formation at sub-MIC drug concentrations.

## Bactericidal drugs induce stochastic fluctuations in population dynamics

Our data above showing the absence of heritable resistance in the surviving populations suggest the possible involvement of stochasticity. That is, bactericidal drugs might induce stochastic fluctuations in the bacterial population size. To investigate this possibility, we performed a plate assay as above and followed the population dynamics of growing micro-colonies at single-cell resolution; we spread cells on agar plates, and rather than waiting for 18 hr and counting colonies visible to the naked eye, we examined how isolated single cells grew to form micro-colonies using time-lapse microscopy. First, as a control, we examined the dynamics in the absence of antibiotics (*Video 1*). We counted the number of growing cells in each colony and plotted the number versus time (*Figure 2A*). The colonies proliferated homogeneously, meaning that different colonies grew similarly. We then repeated the experiment using a bacteriostatic drug (chloramphenicol, thiolutin, or tetracycline) at a sub-MIC level. Visually inspecting the image sequences, we found that cells stably grew, albeit at lower rates (*Video 2*). The number of growing cells in each micro-colony increased homogeneously and similarly (*Figure 2B* and *Figure 2—figure supplement 1*). Previously, we developed a microfluidic chemostat for cell culture (*Kim et al., 2012*; *Deris et al., 2013*). When we repeated these experiments using this device, we again observed the same homogeneous population dynamics (*Figure 2—figure supplement 2A*).

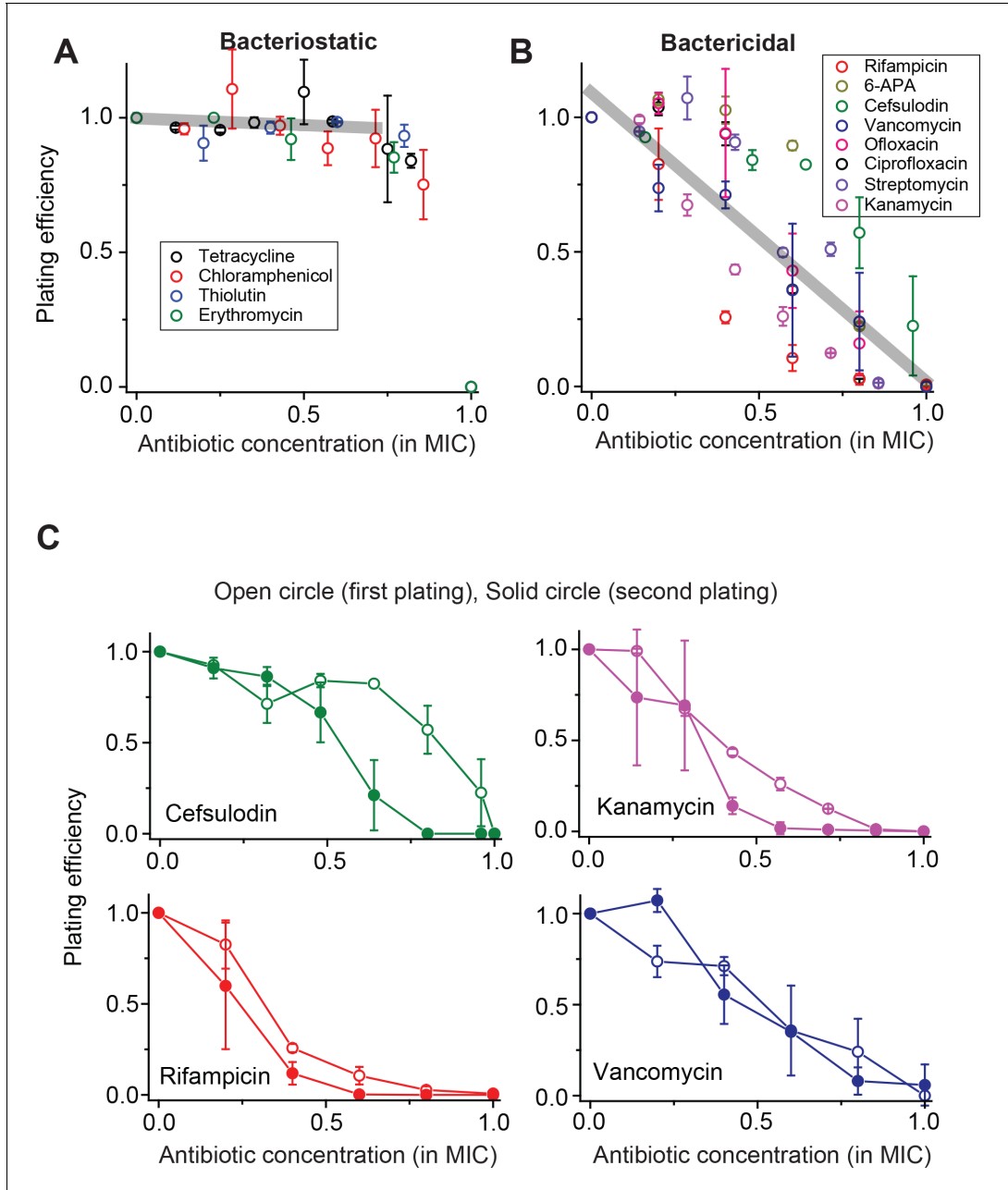

**Figure 1.** Contrasting trends in plating efficiency at increasing concentrations of bacteriostatic and bactericidal drugs. (**A**) When we performed the plate assay using bacteriostatic drugs, $N_{CFU}$ was generally maintained at increasing drug concentrations. See *Figure 1—figure supplement 1* for a detailed illustration of how this plot was made. A linear regression analysis was performed for the whole data set below $0.75 \times MIC$, resulting in the grey line (slope = −0.05, intercept = 0.9942, R-squared = 0.99). Colony size decreased at increasing drug concentrations (*Figure 1—figure supplement 2*). (**B**) For bactericidal drugs, $N_{CFU}$ decreased at increasing drug concentrations, indicating heterogeneous population dynamics of bacteria. A linear regression analysis was performed for the whole data set, and the result was plotted as a grey line (slope = −1.07, intercept = 1.10, R-squared = 0.79). For both groups of drugs, the lowest concentration yielding no colonies was defined as the MIC. The *Supplementary File 1* lists the MICs for all drugs examined in this study. We plotted plating efficiency for each antibiotic in separate panels in *Figure 1—figure supplement 4*. (**C**) For each bactericidal drug used in the experiment (for which the results were shown in *Figure 1B*), we picked a few colonies from the agar plate exhibiting a plating efficiency of ~0.5 (e.g., near $0.6 \times MIC$) and plated them immediately on fresh agar plates containing various concentrations of the same drug. The plating efficiency was similar or marginally lower on the second plating, possibly because exposure to the antibiotics on the first plate adversely affected the cells and rendered them more susceptible to the antibiotics on the second plating. See *Figure 1—figure supplement 3* for similar results for other drugs. Therefore, the ability of bacteria to grow and form colonies on plates containing bactericidal drugs was not heritable. We performed at least two biological repeats for all the experiments and plotted the mean here. The error bars represent one standard deviation from the repeats.
DOI: https://doi.org/10.7554/eLife.32976.002

*Figure 1 continued on next page*

*Figure 1 continued*

The following figure supplements are available for figure 1:

**Figure supplement 1.** We illustrated how we obtained *Figure 1A and B*, using the results for rifampicin as an example.
DOI: https://doi.org/10.7554/eLife.32976.003
**Figure supplement 2.** We spread cells on agar plates containing various concentrations of bacteriostatic drugs, and after 18 hr of incubation, measured the size of the colonies (using ImageJ software).
DOI: https://doi.org/10.7554/eLife.32976.004
**Figure supplement 3.** For each bactericidal drug used (for which the results were shown in *Figure 1B*), we picked a few colonies from an agar plate exhibiting the plating efficiency of ~0.5 (e.g., near 0.6 × MIC) and plated them immediately on fresh agar plates containing various concentrations of the same drug.
DOI: https://doi.org/10.7554/eLife.32976.005
**Figure supplement 4.** We previously plotted plating efficiency for all the antibiotics tested in two panels (*Figure 1A–B*).
DOI: https://doi.org/10.7554/eLife.32976.006

Next, we characterized population dynamics for a bactericidal drug (cefsulodin, ofloxacin, kanamycin, or 6-APA), at a sub-MIC level. We found that the population dynamics were highly stochastic (*Video 3*). Visual inspection of such image sequences indicated that within a given population, some cells were killed stochastically, whereas other cells survived and divided. Such demographic stochasticity would lead to random fluctuations in the population size. Indeed, the number of growing cells in each colony fluctuated randomly over time (*Figure 2C* and *Figure 2—figure supplement 3A–C*). These fluctuations led to dramatically different dynamics for different colonies, even though they originated from genetically identical cells and were cultured under homogeneous antibiotic conditions. When we repeated these experiments using the microfluidic chemostat, we again observed significant population fluctuations (*Figure 2—figure supplement 2B–D*).

Importantly, the fluctuations drove some colonies into extinction (the light red-shaded area in *Figure 2C*). We counted the number of colonies that went extinct and plotted the probability of colony extinction at various drug concentrations; here, we are interested in colony extinction because it is equivalent to bacterial clearance, which means treatment success. *Figure 2—figure supplement 4* showed that the extinction probability increased with increasing drug concentrations. This increase in the extinction probability agrees with the trend of decreasing plating efficiency we found above (*Figure 1B*).

## The effects of bactericidal drugs on cell growth and death

A population will undergo extinction if cells die more frequently than divide. Because bactericidal drugs induce cell death, an increase in extinction probability at higher drug concentrations is expected to be due to an increase in the rate of cell death, $\phi$. Additionally, bactericidal drugs inflict damage on cells (*Belenky et al., 2015*; *Lobritz et al., 2015*). Thus, the rate of cell growth, $\lambda$, might decrease at higher drug concentrations, which could also contribute to colony extinction. We next sought to determine how bactericidal drugs affect $\phi$ and $\lambda$. Previous studies of population growth have shown that at a higher concentration of bactericidal drugs, the 'net growth rate', which is equal to $\lambda - \phi$, decreases (*Regoes et al., 2004*). But, to separately resolve changes in $\phi$ and $\lambda$, the growth and death of cells must be examined at single-cell resolution. We analyzed the single-cell-level image sequences we obtained above (see *Figure 2—figure supplement 5* for details of the

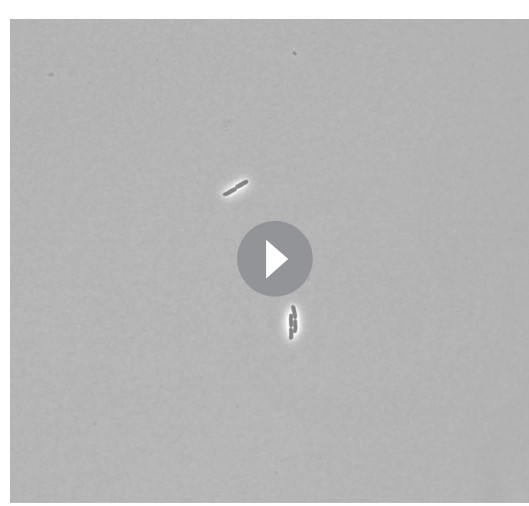

**Video 1.** The growth of micro-colonies in the absence of a drug. We characterized how cells grew and formed micro-colonies on LB agar using time-lapse microscopy. An example image sequence is shown here. The time interval between each frame is 20 min.
DOI: https://doi.org/10.7554/eLife.32976.014

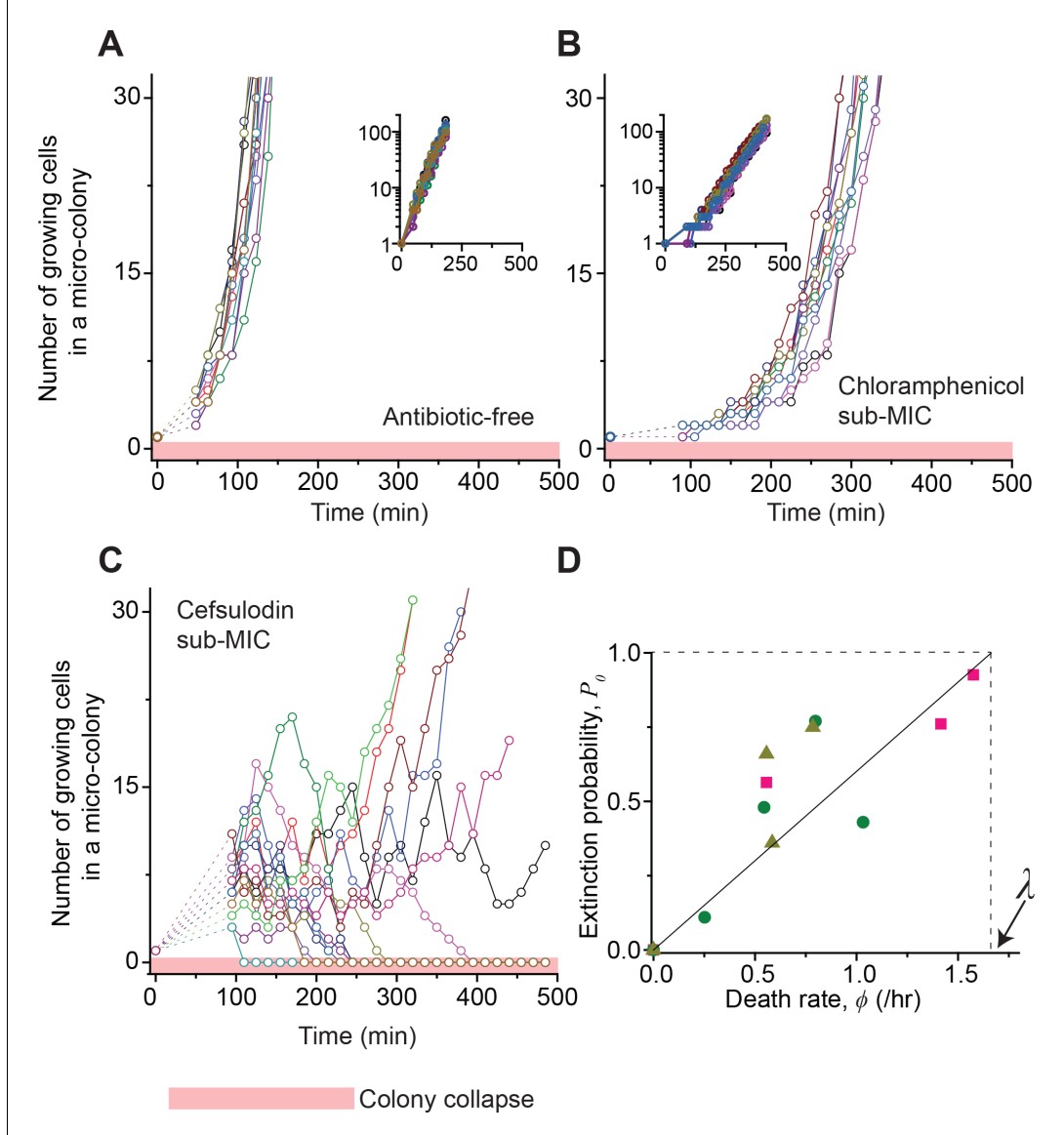

**Figure 2.** Stochastic population dynamics of bacteria exposed to bactericidal drugs. We characterized how cells grow and form micro-colonies on LB agar plates containing different concentrations of antibiotics using time-lapse microscopy. At time zero, we transferred cells growing in antibiotic-free LB liquid medium to a LB agar plate, and confirmed that individual cells were spread out and isolated. Setting up a time-lapse imaging experiment after the transfer took 1 ~ 2 hr, which is why there is a gap in the data immediately after time zero (dashed lines). The experiments were independently repeated twice (biological repeats) and the data from one experiment are shown here. (A, B) We counted the number of growing cells in each micro-colony (represented by a different color). In the absence of antibiotics (panel A) or with a sub-MIC level of a bacteriostatic drug (panel B, $0.7 \times$ MIC of chloramphenicol), the number increased similarly for different colonies, revealing homogeneous population growth. Example image sequences are shown in *Videos 1* and *2*. The data are replotted on a semi-log scale (insets). Such homogeneous population growth was observed for other bacteriostatic drugs (thiolutin and tetracycline) as well; see *Figure 2—figure supplement 1*. When we repeated the experiment using a microfluidic device, we again observed the same homogeneous population dynamics (*Figure 2—figure supplement 2A*). (C) In contrast, the population dynamics of bacteria exposed to a bactericidal drug were highly heterogeneous. An example image sequence was shown in *Video 3*. The number of growing cells within micro-colonies at $0.8 \times$ MIC of cefsulodin is plotted here; the result shows stochastic population fluctuations. Such population fluctuations were again observed when experiments were repeated using other bactericidal drugs (ofloxacin, kanamycin, and 6-APA); see *Figure 2—figure supplement 3A–C*. The light red-shaded region indicates the number equal to zero (i.e., population extinction). When we repeated these experiments using a microfluidic device, we again observed population fluctuations (*Figure 2—figure supplement 2B–D*). (D) Our model predicts that the probability of population extinction increases linearly with death rate, with the slope being $1 / \lambda$ (the solid line). We experimentally characterized the extinction probability (*Figure 2—figure supplement 4*), and the death rate (*Figure 2—figure supplements 5–6*), at different concentrations of bactericidal drugs. Using these data, we obtained the relationship between the extinction probability and the death rate, and plotted it here (green circles: cefsulodin, pink squares: ofloxacin, and grey triangles: 6-APA). We found good agreement between the model prediction and experimental

*Figure 2 continued on next page*

*Figure 2 continued*

data. Note that at increasing drug concentrations, $\lambda$ changed little (*Figure 2—figure supplement 6B*), and thus was taken as a constant in the analysis here.

DOI: https://doi.org/10.7554/eLife.32976.007

The following figure supplements are available for figure 2:

**Figure supplement 1.** We characterized how cells grew and formed micro-colonies at sub-MIC levels of bacteriostatic drugs (chloramphenicol, thiolutin, and tetracycline) at single-cell resolution.

DOI: https://doi.org/10.7554/eLife.32976.008

**Figure supplement 2.** Previously, we developed a microfluidic chemostat for cell culture (*Kim et al., 2012*; *Deris et al., 2013*).

DOI: https://doi.org/10.7554/eLife.32976.009

**Figure supplement 3.** We show population dynamics of bacteria exposed to bactericidal drugs ($0.8 \times$ MIC or $1.2 \times$ MIC).

DOI: https://doi.org/10.7554/eLife.32976.010

**Figure supplement 4.** As discussed in the main text, we analyzed single-cell-level image sequences, and counted the number of colonies that went extinct.

DOI: https://doi.org/10.7554/eLife.32976.011

**Figure supplement 5.** We determined the rates of cell growth $\lambda$ and death $\phi$ at sub-MIC drug concentrations, by analyzing the time-lapse microscopy images of colony growth.

DOI: https://doi.org/10.7554/eLife.32976.012

**Figure supplement 6.** Following the procedure described in *Figure 2—figure supplement 5*, we determined the rates of cell growth $\lambda$ and death $\phi$ at different drug concentrations (green circles: cefsulodin, pink squares: ofloxacin, and grey triangles: 6-APA).

DOI: https://doi.org/10.7554/eLife.32976.013

analysis), and determined $\lambda$ and $\phi$ at various concentrations of bactericidal drugs. We found that at increasing drug concentrations, $\phi$ increased (*Figure 2—figure supplement 6A*). But $\lambda$ changed little, remaining nearly constant (*Figure 2—figure supplement 6B*). Therefore, bactericidal drugs have significant effects on cell death, but not on cell growth.

## A simple, stochastic model of the population dynamics accounts for stochastic clearance of bacterial populations

Our data above suggest that an increase in the probability of population extinction at higher drug concentrations (*Figure 2—figure supplement 4*) is likely due to an increase in the rate of cell death (*Figure 2—figure supplement 6*). To quantitatively understand the relationship between the extinction probability and death rate, we employed a stochastic model, known as the Markovian birth-and-death process, that has been widely used to study the basic features of stochastic population dynamics (*Novozhilov et al., 2006*; *Pavel Krapivsky and Ben-Naim, 2010*; *Kendall, 1948*). This model contains two parameters, the rate of cell growth and death, $\lambda$ and $\phi$, respectively. Each individual cell can divide or die stochastically with the probabilities determined by these parameters. Due to this demographic stochasticity, the number of cells within a population, $n$, fluctuates over time. Thus, $n$ cannot be predicted deterministically but only probabilistically, and the *probability* is described as follows,

$$\dot{P}_n = \lambda(n-1)P_{n-1} - (\lambda+\phi)nP_n + \phi(n+1)P_{n+1}, \quad (1)$$

where $P_n$ refers to the probability of $n$ cells being present in a population. The key boundary condition in this model is that once $n$ reaches 0,

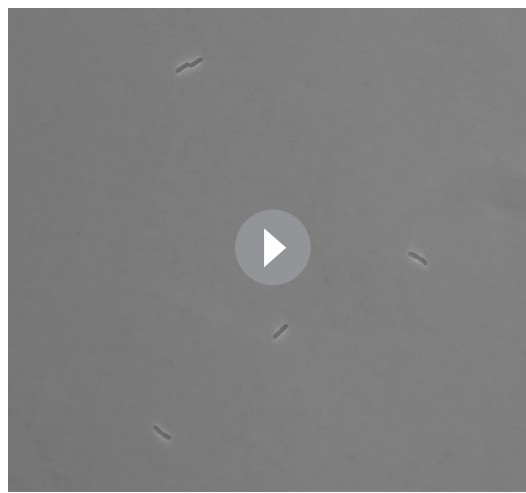

**Video 2.** The growth of micro-colonies with a bacteriostatic drug We characterized how cells grew and formed micro-colonies on LB agar using time-lapse microscopy. $0.7 \times$ MIC of chloramphenicol was used. An example image sequence is shown here. The time interval between each frame is 19 min.

DOI: https://doi.org/10.7554/eLife.32976.015

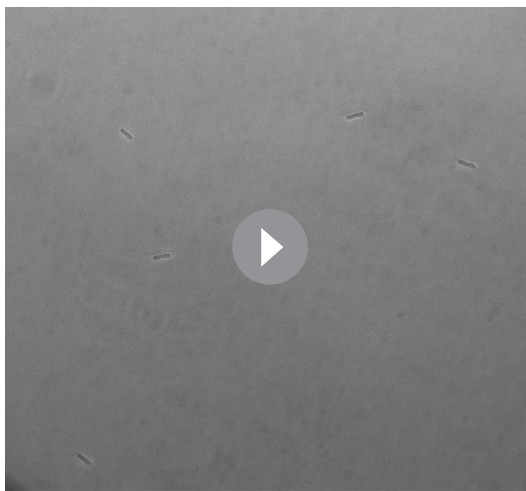

**Video 3.** The growth of micro-colonies with a bactericidal drug We characterized how cells grew and formed micro-colonies on LB agar using time-lapse microscopy. 0.8 × MIC of cefsulodin was used. An example image sequence is shown here. The time interval between each frame is 15 min.
DOI: https://doi.org/10.7554/eLife.32976.016

it cannot change afterward. Known as an 'absorbing boundary', this condition reflects the biological reality that once a population goes extinct, it cannot revive. Therefore, a solution of this model for $n = 0$ (i.e., $P_{n=0}$ or simply $P_0$) describes the dynamics of population extinction. Please see Appendix 1 for the detail and solution.

We first considered the extinction probability $P_0$ at low drug concentrations in which the death rate is lower than the growth rate ($\phi < \lambda$). In this range, $P_0$ is given by their ratio, $P_0 = \phi / \lambda$; see **Equation A6**. Thus, if the death rate is zero ($\phi = 0$), the extinction probability $P_0$ is 0, meaning $n$ always increases (this makes intuitive sense). As the death rate increases ($0 < \phi < \lambda$), $P_0$ increases and becomes non-zero, meaning that $n$ may stochastically reach the absorbing boundary, agreeing with our observation of stochastic population extinction at sub-MIC drug concentrations; **Figure 2C** shows that some populations reached $n = 0$ (marked by the light red-shaded area), while other populations thrived (also see **Figure 2—figure supplement 3A–C**). The solution $P_0 = \phi / \lambda$ predicts that the extinction probability increases linearly with death rate, with the slope being $1 / \lambda$ (the solid line in **Figure 2D**). We sought to test this prediction quantitatively by comparing it with experimental data. Above, analyzing time-lapse microscope images, we obtained the probability of population extinction (**Figure 2—figure supplement 4**), and the death rate (**Figure 2—figure supplement 6**), at different concentrations of bactericidal drugs. Using these data, we obtained the relationship between the probability of population extinction and the death rate, and plotted it in **Figure 2D**. We found good agreement between the model prediction and experimental data (compare the solid line and symbols in **Figure 2D**).

Next, using the quantitative relationship we found above ($P_0 = \phi / \lambda$), we will specify the condition for the MIC. In our plate assay (**Figure 1B**), we observed that the plating efficiency decreases at higher drug concentrations (in the sub-MIC range) and reaches zero at the MIC. Also, the quantitative relationship we found above showed that extinction probability increases at higher drug concentrations (consistent with a decrease in the plating efficiency), reaching one when the death rate is equal to the growth rate (see the dashed line in **Figure 2D**); thus, $P_0 = 1$ at $\phi = \lambda$. The extinction probability equal to one ($P_0 = 1$) means that all colonies go extinct, which corresponds to zero plating efficiency. The drug concentration at which the plating efficiency reaches zero is the MIC (**Figure 1B**). Taken together, at the MIC, the plating efficiency is zero because extinction probability is one ($P_0 = 1$), and the extinction probability is one because the growth rate and death rate are equal to each other ($\phi = \lambda$). In short, $\phi = \lambda$ at the MIC.

Next, we considered drug concentrations above the MIC, where the death rate is higher than the growth rate ($\phi > \lambda$). In this range, the model predicts that all populations eventually go extinct ($P_0 = 1$ in **Equation A6**); this makes intuitive sense. Importantly, due to population fluctuations, populations are expected to go extinct at various times (**Equation A5**), meaning that the number of live populations (the populations that have not undergone extinction yet) decreases gradually over time. The model predicts that this decrease can be approximated by an exponential decay in the long time limit ($t \gg 1/|\phi - \lambda|$); see **Equation A7**. We tested these model predictions by repeating time-lapse microscope experiments at drug concentrations above the MIC. All the populations indeed went extinct at various times (**Figure 3A** and **Figure 2—figure supplement 3D–F**). When we counted the number of live colonies, this number decreased gradually over time (**Figure 3B**). In this semi-log plot, the decrease was linear (compare it with the dashed line), consistent with the model prediction of an exponential decay (**Equation A7**).

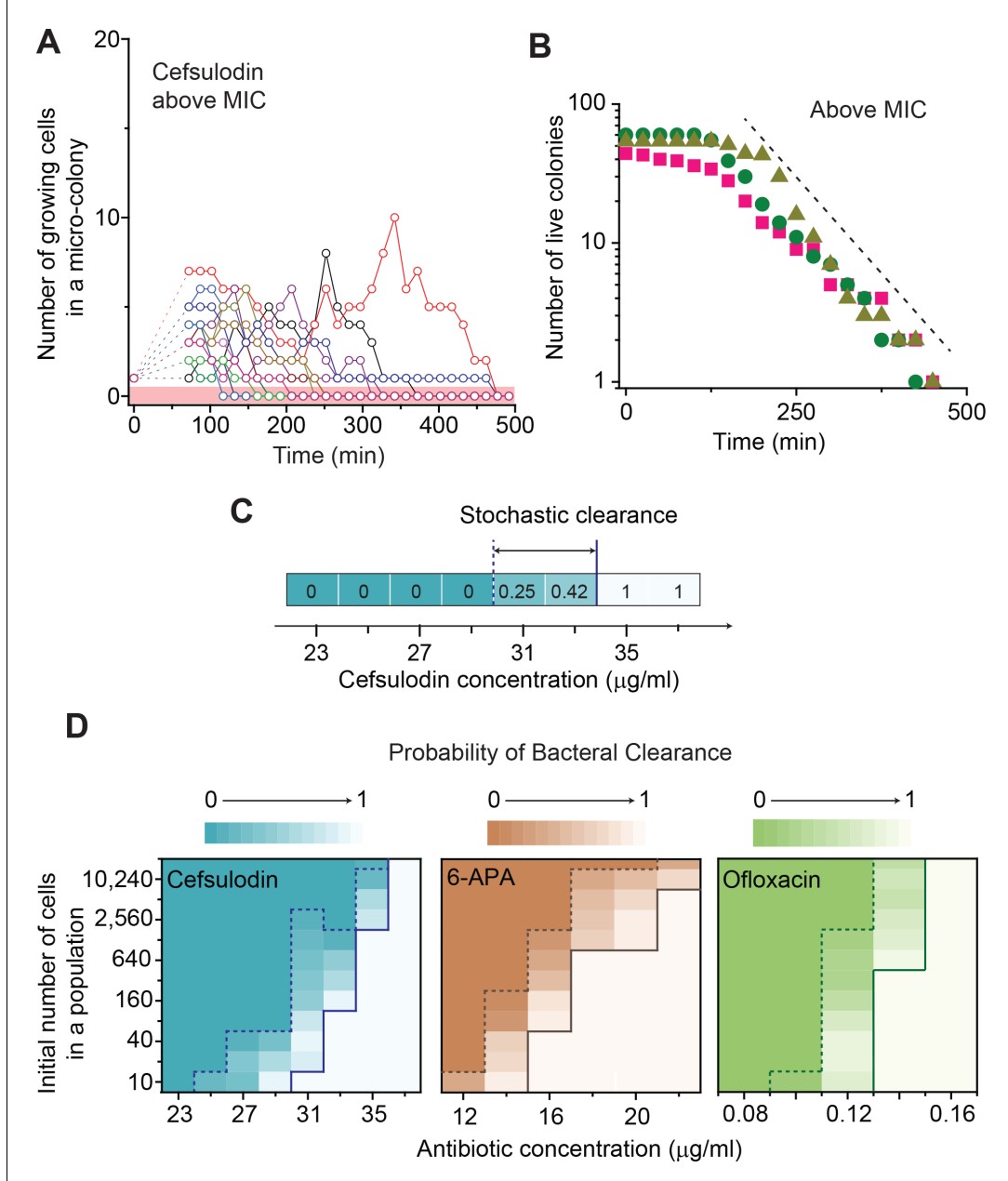

**Figure 3.** Further characterization of stochastic population dynamics. (**A, B**) In *Figure 2C* and *Figure 2—figure supplement 3A–C*, we showed the population dynamics of bacteria exposed to sub-MIC levels of bactericidal drugs. Here, we show the dynamics at drug concentrations above MIC (1.2 × MIC). The number of growing cells within micro-colonies for cefsulodin is plotted in panel A. The results for other bactericidal drugs were plotted in *Figure 2—figure supplement 3D–F*. We observed that all populations went extinct at various times. We then counted the number of live colonies (colonies that have not undergone extinction yet) at various times (~150 colonies monitored). The number decreased gradually over time (green circles: cefsulodin, pink squares: ofloxacin, and grey triangles: 6-APA); see panel B. The decrease was linear in a semi-log plot, consistent with the model prediction of an exponential decay (*Equation A7*). (**C**) We characterized the extinction probability of populations starting with ~640 cells. We prepared a large volume of a cell culture with the cell density of ~640 cells/ 200 μL, supplemented the culture with a low concentration of cefsulodin (23 μg/ml), and then distributed 200 μL of the cell culture equally to 12 isolated chambers in a microtiter plate. We repeated this procedure using higher cefsulodin concentrations (but maintaining the inoculum size). We then incubated the microtiter plate overnight. By counting the chambers that became turbid or clear, we calculated the extinction probability. We used a heat map to graphically represent the probability; for illustration purpose, we also added the values of the probability to the plot. At low cefsulodin concentrations (to the left of the dashed line), all chambers became turbid ($P_0 = 0$). At high concentrations (to the right of the solid line), all chambers were clear ($P_0 = 1$); thus, the solid line indicates MIC. At intermediate concentrations (between the dashed and solid lines), we observed heterogeneous results (only some chambers were clear, $0 < P_0 < 1$). (**D**) We repeated this

*Figure 3 continued on next page*

*Figure 3 continued*

experiment using different inoculum sizes and bactericidal drugs. Please note that although we prepared a large volume culture and distributed it equally to chambers, the number of cells in each chamber might vary. We found that the variation was ~10% or less. See Appendix 3 for details.

DOI: https://doi.org/10.7554/eLife.32976.017

## A population with large inoculum size is subject to stochastic clearance at sub-MIC drug concentrations

Our findings above indicate that the simple stochastic model can adequately capture the extinction dynamics of populations exposed to bactericidal drugs. What is striking in our findings is that, due to drug-induced population fluctuations, a bacterial population may undergo extinction even at sub-MIC concentrations, the concentrations that were previously deemed inefficacious to clear bacteria. We have established this result by examining the dynamics of colonies originated from single bacterial cells, the smallest possible inoculum size. Clinical studies have characterized the bacterial inoculum size that can produce infections (i.e., infectious dose). The infectious dose can be as low as one (*Jones et al., 2006*; *Haas and Rose, 1994*; *Jones et al., 2005*), but is generally 10–100 (*Tuttle et al., 1999*; *DuPont et al., 1989*; *Hara-Kudo and Takatori, 2011*; *Kaiser et al., 1992*) or larger (*Kothary and BABU, 2001*; *Gama et al., 2012*). It is expected that, with larger inoculum size, a population experiences less fluctuations (because demographic stochasticity gets averaged out). When we estimated the magnitudes of fluctuations by the coefficient of variation (CV), that is, the standard deviation divided by the mean, using our model, *Equation A16* shows decreasing CV with increasing inoculum size, supporting the expectation above. Interestingly, *Equation A16* also predicted that the magnitude of population fluctuations depends on rates of cell growth and death as well (*Equation A16*); CV increases as the death rate approaches the growth rate, meaning that population fluctuations become intensified when neither growth nor death is a dominant event. This prediction, together with our finding above that the death rate approaches the growth rate as the drug concentration increases to the MIC (*Figure 2D*), suggests that near the MIC, a population with relatively large inoculum size may still be prone to stochastic extinction.

To test this possibility, we experimentally characterized stochastic clearance of a bacterial population starting with different inoculum sizes. First, we prepared a large volume of a cell culture with the cell density of ~640 cells/ 200 μL, supplemented the culture with a low concentration of cefsulodin (23 μg/ml), and then distributed 200 μL of the cell culture equally to 12 isolated chambers in a microtiter plate. Here, an isolated cell culture in each chamber represents a separate population. We repeated this procedure using higher cefsulodin concentrations (but maintaining the inoculum size). We then incubated the microtiter plate overnight. We found that, at low cefsulodin concentrations (to the left of the dashed line in *Figure 3C*), all chambers became turbid, meaning all populations grew. Thus, the probability of population extinction was zero ($P_0 = 0$). In *Figure 3C*, we used a heat map to graphically represent the probability (we also added the values of the probability in the graph for additional clarification). At high concentrations (to the right of the solid line), all chambers were clear ($P_0 = 1$); thus, the solid line indicates the MIC. At intermediate concentrations (between the dashed and solid lines), we observed heterogeneous population growth; some chambers were clear while others were turbid ($0 < P_0 < 1$). Subsequent plating of the clear cultures on drug-free LB agar plates yielded no colonies, indicating population extinction.

We then repeated this experiment by using different inoculum sizes and bactericidal drugs. The results were plotted as heat maps in *Figure 3D*; here, the Y axis represents different inoculum sizes. As above, a solid line represents the MIC, above which all chambers were clear ($P_0 = 1$). MIC values were higher at higher cell densities. We note that the higher MIC at higher cell density (inoculum effect) has been observed previously and is being actively studied by others in the field (*Artemova et al., 2015*; *Brook, 1989*; *Tan et al., 2012*; *Karslake et al., 2016*). Thus, it is not the focus of our study; rather, we focus on stochastic clearance below the MIC. As above, a dashed line represents the concentrations below which all chambers were turbid ($P_0 = 0$). The area between the dashed and solid lines indicates the range of drug concentrations and inoculum sizes that exhibited heterogeneous population growth ($0 < P_0 < 1$), meaning stochastic clearance. *Figure 3D* shows that stochastic clearance occurs even for a population starting with as large as ~20,000 cells, inoculum size much larger than infectious doses for many infectious diseases.

## Alteration of the extinction probability to facilitate bacterial eradication at sub-MIC drug concentrations

Previously, antibiotic treatment at sub-MIC levels was not considered a viable option for bacterial eradication, because the deterministic model predicts that all bacterial populations should grow at sub-MIC levels (i.e., antibiotic treatment failure). However, our experimental results and stochastic model above indicate that at sub-MIC levels of bactericidal drugs, a population might undergo extinction stochastically. An increase in this probability while keeping the drug concentration low would be therapeutically useful; with an increased probability of extinction, sub-MIC ranges of drugs could be used to eradicate bacteria reliably. We therefore employed our model to explore how the extinction probability can be altered by means other than changing the bactericidal drug concentration.

Our model indicates that the extinction probability is determined by the ratio of the death and growth rates ($P_0 = \phi / \lambda$; *Equation A6*). Thus, based on the model, a reduction in growth rate (denominator) should lead to an increase in the extinction probability. Growth rate can be reduced by using poor growth media, or alternatively using bacteriostatic drugs. This means, for a sub-MIC concentration of a bactericidal drug (for which the extinction probability is less than 1), either a switch to poor growth media or addition of a sub-MIC level of a bacteriostatic drug would lead to an increase in the extinction probability. We note that the latter represents combination therapies, and other studies have characterized bacterial responses to combination therapies (*Bollenbach, 2015*). However, these studies primarily concerned deterministic changes in the MIC of a large population. Conversely, our study focuses on how combination therapies affect stochastic occurrence of population extinction. Another difference is our focus on sub-MIC drug ranges, an important point given previous research showing that the effects of drug combinations at the MIC might differ from those at sub-MIC levels (*Ocampo et al., 2014*).

To characterize the extinction probability, we introduced the plating inefficiency (= 1 – plating efficiency); the plating efficiency reflects the probability that a bacterial cell forms a population of a bacterial colony, and therefore, the plating inefficiency reflects the probability of population extinction. We first calculated the plating inefficiency using the plate assay results in *Figure 1B* and plotted it as white columns in *Figure 4A*; thus, white columns represent the extinction probability in rich growth media (LB) with bactericidal drugs alone. To test the effect of our growth reduction strategy, we repeated the plate assay, either by replacing LB with casamino acids (a poor nutrient source that leads to slower growth than LB; see *Figure 4—figure supplement 1*), or by adding a sub-MIC concentration of chloramphenicol or tetracycline. The plating inefficiency obtained with these treatments was plotted as solid columns in *Figure 4A*. The rise of solid columns above white columns indicates that growth reduction indeed led to an increase in the extinction probability, in agreement with our prediction.

We next examined how generally such an increase in the extinction probability might occur. The extinction probability depends on the growth rate ($\lambda$) and death rate ($\phi$), which we assumed to be independent. Because the probability is equal to their ratio ($\phi/\lambda$; *Equation A6*), this growth reduction strategy might not work when $\phi$ is not independent but decreases in response to a decrease in $\lambda$. Such coupling between $\phi$ and $\lambda$ could occur for bactericidal drugs that kill only growing cells, possibly because these drugs target processes critical for cell growth. This means that, for bactericidal drugs that exhibit a killing rate of zero for non-growing cells (i.e., $\phi \to 0$ when $\lambda \to 0$), neither the switch to poor growth medium nor addition of bacteriostatic drugs would increase the extinction probability. To test this possibility, we first identified such bactericidal drugs; we stopped cell growth in cultures by depriving the cells of nutrients, added bactericidal drugs at concentrations capable of eradicating growing cells, and then determined which drugs were no longer capable of killing the bacteria. As shown in *Figure 4—figure supplement 2*, killing was completely abolished for kanamycin and 6-APA, indicating that $\phi \to 0$ when $\lambda \to 0$. As expected, for these drugs, neither the switch to poor growth medium nor addition of bacteriostatic drugs resulted in an increase in the extinction probability (*Figure 4—figure supplement 3*).

Furthermore, our model does not consider specific drug-drug interactions. For example, a previous study showed bacteriostatic translation-inhibiting drugs and bactericidal quinolone drugs affect gene expression in a way to negate their effects (*Bollenbach et al., 2009*). Consistent with this study, we failed to observe significant changes in the extinction probability when ofloxacin or

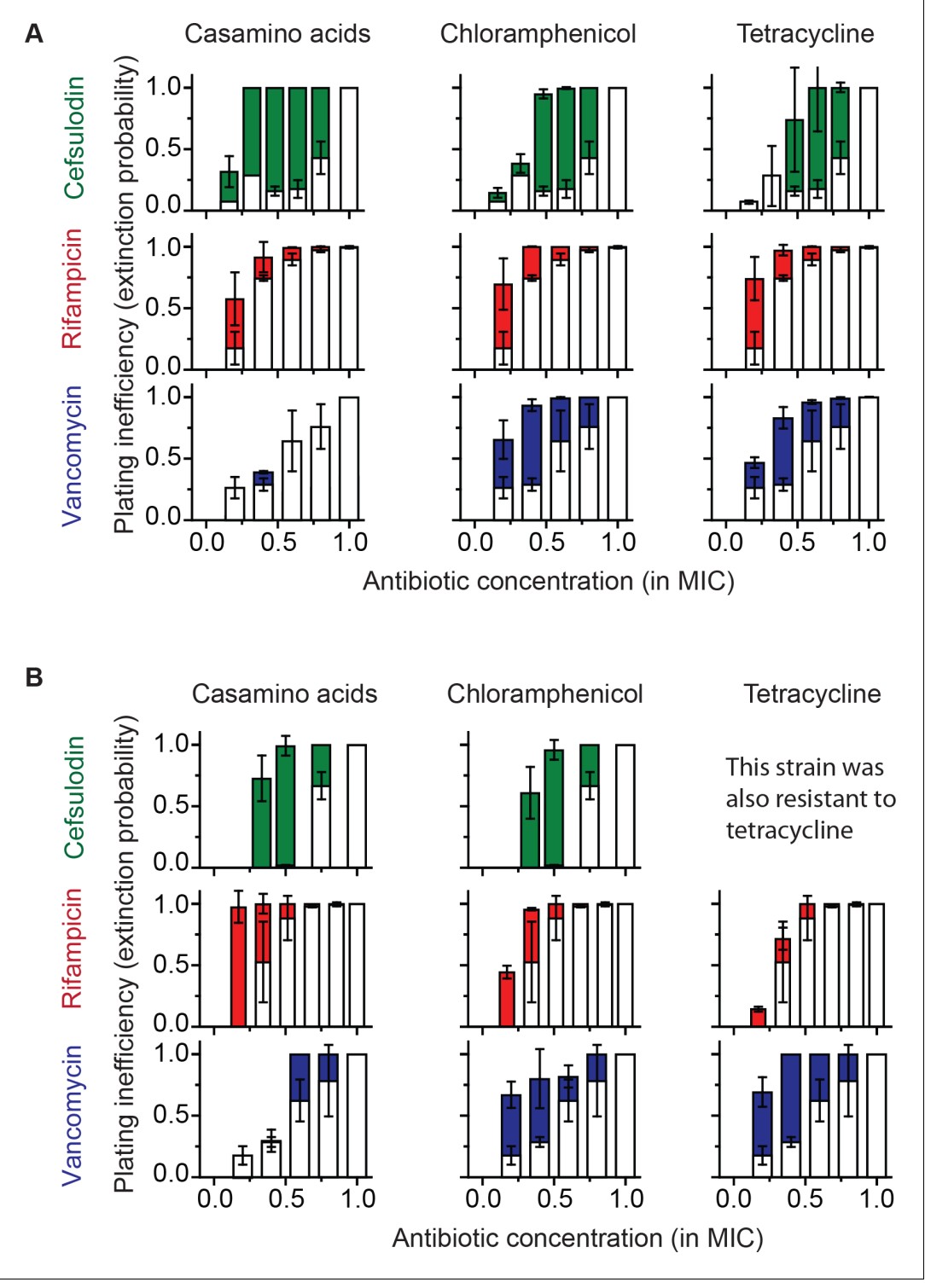

**Figure 4.** Increasing the population-extinction probability by a growth reduction strategy. (**A, B**) Previously, we plated WT (antibiotic-susceptible) *E. coli* cells on LB agar plates at various concentrations of cefsulodin and plotted the plating efficiency in **Figure 1B**. Here, we plotted the plating inefficiency (1- plating efficiency) as white columns in the top row; the plating inefficiency represents the extinction probability. We then repeated a plate assay, either by using casamino acids agar plates (left), or by adding a sub-MIC concentration of a bacteriostatic drug, chloramphenicol (0.7 × MIC, middle) or tetracycline (0.5 × MIC, right), to LB agar plates. Note that casamino acids lead to slower growth than LB; see **Figure 4—figure supplement 1**. We plotted the plating inefficiency

*Figure 4 continued on next page*

*Figure 4 continued*

obtained with these treatments as solid columns in panel A. We repeated this procedure for rifampicin (middle row) and vancomycin (bottom row), for kanamycin and 6-APA (*Figure 4—figure supplement 3*), and for ciprofloxacin and ofloxacin (*Figure 4—figure supplement 4*). Note that how generally growth reduction leads to an increase in the extinction probability depends on whether $\phi$ is dependent on $\lambda$ or not; see the main text and *Figure 4—figure supplement 2*. We then used antibiotic-resistant strains and repeated these experiments (panel B). Please note that the MICs of these mutants were five to ten fold higher than those of the WT strain. The rise of solid columns above white columns indicates an increase in the extinction probability. We performed at least two biological repeats for all the experiments and plotted the mean. The error bars represent one standard deviation from the repeats.

DOI: https://doi.org/10.7554/eLife.32976.018

The following figure supplements are available for figure 4:

**Figure supplement 1.** We cultured cells in either LB medium or N-C- medium supplemented with 2% of casamino acids (*Csonka et al., 1994*).
DOI: https://doi.org/10.7554/eLife.32976.019

**Figure supplement 2.** We measured MICs of bactericidal drugs using exponentially-growing cultures and confirmed that at 1.5 × MICs, all growing cells were eradicated; that is, survival fraction (SF) was equal to zero.
DOI: https://doi.org/10.7554/eLife.32976.020

**Figure supplement 3.** In *Figure 4—figure supplement 2*, we found that kanamycin and 6-APA do not kill non-growing cells (i.e., $\phi \to 0$ when $\lambda \to 0$).
DOI: https://doi.org/10.7554/eLife.32976.021

**Figure supplement 4.** A previous study showed bacteriostatic translation-inhibiting drugs and bactericidal quinolone drugs affect gene expression in a way to negate their effects (*Bollenbach et al., 2009*).
DOI: https://doi.org/10.7554/eLife.32976.022

ciprofloxacin (quinolone drugs) is used with and without bacteriostatic translation-inhibiting drugs (*Figure 4—figure supplement 4*).

## Extending the growth-reduction strategy to antibiotic-resistant strains

Above, we tested the growth reduction strategy for a WT (antibiotic-susceptible) *E. coli* strain. Although the strategy did not work for some drugs due to their complex effects on cells, for those that worked, the strategy substantially increased the chance of bacterial clearance at sub-MIC drug concentrations. We wondered if this strategy could be applicable to antibiotic-resistant strains. Resistant strains are difficult to eradicate because their MICs are very high, to levels that are toxic to hosts. Therefore, the development of therapies that utilize sub-MIC doses would be highly useful. To test if our growth-reduction strategy would work for antibiotic-resistant bacteria, we repeated a plate assay using resistant strains. Laboratory evolution of rifampicin resistance has been frequently reported in the literature (*Goldstein, 2014*). By plating WT (antibiotic-susceptible) *E. coli* cells on a LB agar plate containing the rifampicin concentration equal to 2 × MIC, we isolated a rifampicin-resistant (Rif$^r$) mutant; the MIC of this mutant was ~10 fold higher than that of the parent strain. We then plated the mutant on LB agar plates containing various rifampicin concentrations. Additionally, we acquired clinically-isolated *E. coli* strains that were resistant to either cefsulodin or vancomycin (see Methods), and plated them on LB agar plates with increasing concentrations of cefsulodin or vancomycin. These resistant strains exhibited non-zero plating inefficiency at sub-MIC concentrations of bactericidal drugs (white columns in *Figure 4B*), indicating stochastic clearance. We then repeated the experiments either by replacing LB with casamino acids or by adding a sub-MIC concentration of a bacteriostatic drug. As with the antibiotic-susceptible strain, these treatments led to an increase in the plating inefficiency (*Figure 4B*), indicating that our growth reduction strategy facilitated the clearance of antibiotic-resistant bacteria at sub-MIC concentrations.

## Discussion

Antibiotic treatment typically targets mature infection which contains a large number of bacterial cells (e.g., $\geq 10^8$) (*Smith and Wood, 1956*; *Palaci et al., 2007*; *Feldman, 1976*; *Canetti, 1965*; *Canetti, 1956*). To clear infections and avoid post-treatment relapse, not only the reduction of a large population of bacteria to a small population, but also the complete extinction of the small

population is desired (*Tomita et al., 2002*; *Wilson et al., 2013*; *Bayston et al., 2007*); this is especially so for immuno-compromised hosts and also for infections involving bacteria with a low infectious dose. Previous studies of large bacterial populations adequately accounted for the former process. This study focused on the latter process. Our results directly revealed that antibiotics induce significant fluctuations in population size, leading to stochastic population extinction. Modeling population fluctuations using a probabilistic model, we then established a quantitative understanding of stochastic extinction. This model further predicted how the extinction probability could be manipulated to facilitate bacterial eradication at sub-MIC drug concentrations. We experimentally tested how amenable the extinction probability is to manipulation.

One possible molecular-level mechanism that gives rise to population fluctuations could be cell-to-cell variability in gene expression. Previous studies showed that variation in expression of antibiotic-resistance genes, *marA*, *cat*, *kagG*, *ompC* or *bla*, results in variation in antibiotic susceptibility to carbenicillin, chloramphenicol, isoniazid, kanamycin or ceftriaxone, respectively (*Deris et al., 2013*; *El Meouche et al., 2016*; *Wakamoto et al., 2013*; *SanchezSánchez-Romero and Casadesus, 2014*; *Wang et al., 2014*). Although our study mostly focuses on antibiotic-susceptible bacteria, a similar mechanism might play a role, leading to heterogeneous growth/death of bacterial cells and eventually population fluctuations. We note that there were attempts to stochastically model large populations of antibiotic-susceptible bacteria (e.g., see [*Ferrante et al., 2005*]). But, given the deterministic nature of observed dynamics, the need for stochastic models was not clear, and the model prediction of stochasticity was not tested in the work. On the other hand, previous theoretical studies of the evolution of antibiotic resistance typically modeled the growth and death of newly-emerged mutants as stochastic processes, showing how fluctuations in the size of small mutant populations affect evolutionary dynamics; e.g., see (*Hermsen et al., 2012*; *Nissen-Meyer, 1966*). Our study validates this modeling approach.

Our findings expose fundamental limits in our predictive ability for bacterial clearance. Clinical studies of antibiotic therapies have often reported unexpected failures of eradicating antibiotic-susceptible bacteria (*Doern and Brecher, 2011*; *Weidner et al., 1999*; *Gopal et al., 1976*; *Ficnar et al., 1997*; *Forrest et al., 1993*). Laboratory studies of simple model organisms such as worms have reported similar observations (*Moy et al., 2006*; *Needham et al., 2004*; *Kaito et al., 2002*). The variability in host environments could certainly contribute to such unexpected antibiotic failures. For example, a recent study showed the effects of variability in host immunity on infection course (*Duneau et al., 2017*). Our study demonstrates that even in the absence of host variability, bacterial clearance occurs stochastically due to antibiotic-induced population fluctuations. At sub-MIC drug concentrations, bacterial populations may or may not go extinct. At drug concentrations above the MIC, all populations eventually go extinct, but not all at once. Rather, the extinction time is highly variable, meaning that in some cases, it can take significantly longer to eradicate bacteria. This inherent stochasticity, together with host variability, makes it difficult (or even impossible) to deterministically predict antibiotic-mediated clearance of bacterial infection.

More studies are needed to elucidate the impact of these population fluctuations on treatment outcomes in clinical settings. However, we observed such fluctuations even for a relatively large population ($\sim 10^4$ cells). In comparison, previous in vivo studies showed that the population size needed to establish infections (i.e., infectious dose) can be as small as 1–100 cells (*Jones et al., 2006*; *Haas and Rose, 1994*; *Jones et al., 2005*; *Tuttle et al., 1999*; *DuPont et al., 1989*; *Hara-Kudo and Takatori, 2011*; *Kaiser et al., 1992*), which is well within the stochasticity range. This means that if tens of cells (or even a few cells) happen to stochastically survive a fixed course of antibiotic treatment, this small population can re-establish infections once antibiotics are removed, leading to treatment failure. Importantly, a recent article raised an issue regarding the conventional wisdom of 'complete the prescribed course', and argued for re-consideration of antibiotic duration (*Llewelyn et al., 2017*). We believe that our observation of stochastic extinction dynamics, especially inherent variability in extinction time, has significant bearing on this issue.

Furthermore, our study may guide the design of new therapeutic strategies. Based on the deterministic understanding of population dynamics, it has been generally accepted that only at drug concentrations above the MIC, bacterial populations go extinct. Contrarily, we observed that stochastic population fluctuations drive a population to extinction even at sub-MIC drug concentrations. This observation suggests an intriguing possibility that sub-MIC drug ranges can be used as a clinical option to clear bacteria. We acknowledge that the stochastic nature in the population dynamics is a

double-edged sword. Stochasticity can be advantageous because it can drive a population to extinction even at low drug concentrations. But, stochasticity makes it impossible to pre-determine whether the bacterial population of particular interest will go extinct or not; this is disadvantageous because we cannot predict *a priori* if a specific treatment will work or not. However, this disadvantage can be minimized by manipulating population fluctuations and thereby increasing the probability of extinction. In the present study, we explored this possibility with the help of the probabilistic model, showing that it is possible to increase the probability of clearance. This idea of using sub-MIC drug concentrations to clear bacteria is particularly attractive in the context of antibiotic resistance. Antibiotic-resistant bacteria have very high MICs, often above the levels that are toxic to hosts. Thus, antibiotic concentrations above the MIC cannot be administered, which is why antibiotic resistance is a serious public concern worldwide (*O'Neill, 2016*). Our plate assay using bactericidal drugs revealed that, like antibiotic-susceptible bacteria, resistant bacteria exhibited non-zero plating inefficiency below the MIC, indicating a non-zero probability of clearance at sub-MIC drug concentrations. We also showed that this probability of clearance could be manipulated, facilitating the clearance of antibiotic-resistant bacteria. For comparison, we note a recent study in the field of viral infections, which showed that stochastic noise in HIV gene expression may be used to treat HIV infections (*Dar et al., 2014*). This study further supports the idea that stochasticity can be advantageous and be used to combat infections. We believe that the time is ripe for the development of clinical treatment strategies to take advantage of stochasticity, especially so given recent advances in our understanding of stochasticity in biological processes (*Jones et al., 2014*; *Ackermann, 2015*; *Tanouchi et al., 2015*; *Banerjee et al., 2017*; *Scott et al., 2007*; *Schmiedel et al., 2015*; *Sigal et al., 2006*; *Blount et al., 2008*; *Ray and Igoshin, 2012*).

Our study will also have positive impacts on *in vitro* assessment of antibiotic efficacy. MIC is the most critical parameter to assess antibiotic efficacy. In the deterministic framework, MIC is defined as the drug concentration at which the population size is maintained, which is realized when the death rate is equal to growth rate ($\phi = \lambda$); see *Equation A2*. Accordingly, in a broth dilution method, the drug concentration that yields no change in the turbidity of bacterial cultures is defined as the MIC. Above, we found that in the stochastic framework, at the MIC, the rates of death and growth are equal ($\phi = \lambda$; see *Figure 2D*) and the extinction probability is 1 (*Equation A6*), meaning all populations go extinct at the MIC. Therefore, in both deterministic and stochastic frameworks, at the MIC, the condition, $\phi = \lambda$, is satisfied, but population dynamics are very different (population maintenance versus population extinction). This clarification can reconcile two common ways to determine the MIC, a plate assay based on complete colony extinction and a broth method based on no change in culture turbidity (population maintenance); although the MICs were determined based on different population dynamics in these two cases, both methods identify the drug concentration at which growth rate is equal to death as the MIC. This clarification is particularly important in light of recent efforts to increase the efficiency of the broth method by using small culture volumes (which include a few or tens of bacterial cells) (*Avesar et al., 2017*). With such small volumes, MIC should be defined based on population extinction, not population maintenance.

Lastly, our findings have implications on bacterial persistence. Dormant cells are refractory to antibiotics, persisting through antibiotic treatments (*Allison et al., 2011*; *Balaban et al., 2004*). They are present in very low frequencies (typically $10^{-5}$, meaning 1 out of $10^5$ cells) (*Lewis, 2010*), and thus have little effects on population dynamics in small populations considered here ($<<10^5$). However, a study of persister formation requires the enrichment of persisters. To enrich them, studies often treat a large population using antibiotics and characterize a small population of survivors as persisters. The inherently stochastic nature of a small population may lead to variability in this process of enrichment and characterization, complicating studies of persisters. In fact, such variability was reported by a recent quantitative study of persistence (*Brauner et al., 2017*). Therefore, our findings on the dynamics of small populations could be useful for a better understanding of persistence.

## Materials and methods

### Bacterial strains and culture

Experiments were conducted using *E. coli* strain NCM3722 (*Soupene et al., 2003*; *Lyons et al., 2011*; *Brown and Jun, 2015*). Bacteria were grown in 5 mL of Lysogeny Broth (LB, Fisher Bioreagents) in 20 mL borosilicate glass culture tubes at 37° C with shaking (250 rpm). Our typical experimental procedure is as follows. Cells were first cultured in LB broth overnight (pre-culture). The next morning, the cells were sub-cultured into pre-warmed LB broth at the optical density ($OD_{600}$) of ~0.001 (measured using a Genesys20 spectrophotometer, Thermo-Fisher) and allowed to grow exponentially. The culture at the $OD_{600}$ of ~0.4 was used for a plate assay or for microscope experiments.

We evolved a rifampicin-resistant mutant from our WT strain (NCM3722). We plated ~$10^8$ WT cells on a LB agar plate containing 20 µg/ml ($2 \times MIC^{WT}$). 18 colonies were formed next day. We chose one colony and purified it by re-streaking. The mutant (EMK32) had the MIC of 160 µg/ml. We obtained the cefsulodin-resistant strain (EMK35, MIC = 170 µg/ml) and vancomycin-resistant strain (EMK36, MIC = 250 µg/ml) from Georgia Emerging Infections Program MuGSI collection (their MuGSI strain numbers were Mu519 and Mu107, respectively.)

### Plate assay

Cells were spread on LB agar plates containing different concentrations of various antibiotics (see below). Through serial dilutions (using 1.16% (w/v) NaCl solution), we ensured the number of colonies to be between 50 and 250 on a plate ($100 \times 15$ mm Petri dish). The plates were then incubated at 37°C. After 18 hr of incubation, the number of visible colonies on plates was determined. As indicated in the main text, we also used casamino acids agar plates. We dissolved 2% casamino acids in N-C- minimal medium (*Csonka et al., 1994*) and filtered the medium. We separately autoclaved agar, and when agar was cooled and felt warm to the touch, the casamino acids medium was added.

### Antibiotics

Stock solutions of ciprofloxacin (1 mg/ml), kanamycin (50 mg/ml), 6-APA (2 mg/ml), streptomycin (25 mg/ml), ofloxacin (1 mg/ml), erythromycin (10 mg/ml), vancomycin (100 mg/ml), and cefsulodin (30 mg/ml) were prepared in sterilized water. Stock solutions of tetracycline (50 mg/ml) and chloramphenicol (10 mM) were prepared in methanol. Stock solutions of thiolutin (2 mg/ml) and rifampicin (50 mg/ml) were prepared in DMSO. Antibiotics were purchased from Biobasic Inc (Canada), Acros Organics, or Sigma-Aldrich (St. Louis, MO.).

### Time-Lapse microscopy

Cells were cultured as described above without antibiotics first. At $OD_{600}$ of ~0.4, 2 µL of aliquot from a culture was loaded into a pre-warmed 35 mm glass-bottom Petri dish (InVitro Scientific). A pre-warmed LB agarose pad containing antibiotics was placed over them (this procedure marked time zero). The dish was then moved into a pre-warmed (at 37°C) inverted microscope (Olympus IX83), and appropriate stage positions (typically ~50 positions) were selected. Selection of multiple state positions and setting up the software for automatic image acquisition normally took 1 ~ 2 hr. The microscope had an automated mechanical XY stage and auto-focus, and was controlled by the MetaMorph software (Molecular Devices). Also, it was housed by a microscope incubator (InVivo Scientific) which maintained the temperature of samples at 37°C during experiments. An oil immersion phase-contrast $60 \times$ objective was used for imaging. Images were captured using a Neo 5.5 sCMOS camera (Andor). We also cultured cells in a microfluidic chemostat (using LB broth as growth medium). The detailed procedure for the microfluidic experiments was published in our previous articles (*Kim et al., 2012*; *Deris et al., 2013*). Images were analyzed using MicrobeJ, a plug-in for the ImageJ software (*Ducret et al., 2016*), and the analysis results were validated manually. In our experiments, all cells that stopped growing eventually underwent lysis. Although dormant cells that survive antibiotics (i.e., persisters) may complicate our analysis, their frequency is very low, ~$10^{-5}$ (i. e., 1 in $10^5$ cells) (*Lewis, 2010*). Because our study involves small populations (less than 100 cells), dormancy has little relevance to our study.

## Replicate culture using a microtiter plate

We first cultured cells in 5 mL LB medium to the $OD_{600}$ of ~0.4, as described above. Previously, using a plate assay, we determined that $OD_{600}$ of 0.4 contains $2 \times 10^8$ cells per 1 mL. Using this relation, we diluted the culture in 10 mL LB broth such that there were ~640 cells per 200 μL (we separately confirmed this density by plating cells on LB agar plates containing no antibiotics and counting colony-forming units). We supplemented this culture with various concentrations of antibiotics. We then equally distributed 200 μL of this culture to 12 chambers in a microtiter plate. We repeated this procedure by varying cell density. Next day, we measured the $OD_{600}$ of each chamber.

## Acknowledgement

We thank Jeff Gore, Véronique Perrot, and Bruce Levin for helpful comments on the manuscript. We also thank Emily Crispell and David Weiss for sharing strains with us. This work was funded by Research Corporation for Science Advancement (#24097) and the Human Frontier Science Program (RGY0072/2015).

## Additional information

### Funding

| Funder | Grant reference number | Author |
|---|---|---|
| Research Corporation for Science Advancement | 24097 | Jessica Coates<br>Bo Ryoung Park<br>Dai Le<br>Emrah Şimşek<br>Minsu Kim |
| Human Frontier Science Program | RGY0072/2015 | Jessica Coates<br>Bo Ryoung Park<br>Dai Le<br>Emrah Şimşek<br>Minsu Kim |

The funders had no role in study design, data collection and interpretation, or the decision to submit the work for publication.

### Author contributions

Jessica Coates, Bo Ryoung Park, Dai Le, Emrah Şimşek, Validation, Investigation, Methodology; Waqas Chaudhry, Visualization; Minsu Kim, Conceptualization, Resources, Formal analysis, Supervision, Funding acquisition, Investigation, Methodology, Writing—original draft, Project administration, Writing—review and editing

### Author ORCIDs

Minsu Kim (iD) http://orcid.org/0000-0003-1594-4971

### Decision letter and Author response

Decision letter https://doi.org/10.7554/eLife.32976.031
Author response https://doi.org/10.7554/eLife.32976.032

## Additional files

### Supplementary files

• Supplementary file 1. Minimum inhibitory concentration (MIC) for all drugs examined in this study
Source data: Source data for all figures
DOI: https://doi.org/10.7554/eLife.32976.023

• Transparent reporting form
DOI: https://doi.org/10.7554/eLife.32976.024

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

## Appendix 1

DOI: https://doi.org/10.7554/eLife.32976.025

### A deterministic model of population dynamics

If cells replicate at rate, $\lambda$, and die at rate, $\phi$, a general deterministic model of population growth can be described by

$$\frac{dn}{dt} = \lambda \cdot n - \phi \cdot n, \tag{A1}$$

where $n$ is the number of cells in the population. Its solution is

$$n(t) = n(0) \cdot e^{(\lambda - \phi) \cdot t}. \tag{A2}$$

Previous studies modeled the population dynamics of bacteria exposed to antibiotics using this solution; e.g., see (*Regoes et al., 2004*; *Czock et al., 2009*). At drug concentrations below the MIC (sub-MIC), growth rate is higher than death rate ($\lambda > \phi$); thus, a bacterial population grows. When the drug concentration reaches the MIC, growth rate and death rate are equal ($\lambda = \phi$); thus, the population size is maintained. At drugs concentrations above the MIC, growth rate is lower than death rate ($\lambda < \phi$); thus, a bacterial population declines.

### A stochastic model of population dynamics

We found that bactericidal drugs induce demographic stochasticity (i.e., cells may replicate or die stochastically), and thus the population size fluctuates over time. To account for such fluctuations, we use the Markovian birth-and-death process model, which represents the most fundamental formulation of stochastic population growth (*Novozhilov et al., 2006*; *Pavel Krapivsky and Ben-Naim, 2010*; *Kendall, 1948*) and consider the probability of there being $n$ cells at time $t$ in a population, $P_n(t)$. Temporal evolution of $P_n(t)$ is given by the following master equation,

$$\dot{P}_n = \lambda(n-1)P_{n-1} - (\lambda + \phi)nP_n + \phi(n+1)P_{n+1}. \tag{A3}$$

This equation can be solved analytically; see (*Novozhilov et al., 2006*; *Pavel Krapivsky and Ben-Naim, 2010*; *Kendall, 1948*). For a population that originates from a single cell (e.g., a colony on agar originates from a single bacterium in our experiments), the initial condition is $P_1(0)=1$. Then, the solution is

$$P_n(t) = A(t) \cdot a(t)^{n-1}, \tag{A4}$$

where $A(t) = E(t) \cdot \left(\frac{1 - \phi/\lambda}{E(t) - \phi/\lambda}\right)^2$, $a(t) = \left(\frac{E(t) - 1}{E(t) - \phi/\lambda}\right)$, and $E(t) = e^{(\lambda - \phi) \cdot t}$.

For $n = 0$,

$$P_0(t) = \frac{\phi}{\lambda}\left(\frac{E(t) - 1}{E(t) - \phi/\lambda}\right), \tag{A5}$$

which represents the probability that a population has gone extinct before time $t$.

In the long-time limit, $P_0(t)$ converges to

$$P_0 = \begin{cases} 1, & \text{if } \phi \geq \lambda \\ \frac{\phi}{\lambda}, & \text{if } \phi < \lambda \end{cases}. \tag{A6}$$

Because bactericidal drugs induce cell death, the death rate $\phi$ can be increased by using higher concentrations of drugs (*Figure 2—figure supplement 6A*). The solution (*Equation A6*) indicates that, if $\phi$ is equal to or greater than $\lambda$, a population will always go extinct, meaning that the lowest value of $\phi$ that can consistently achieve population extinction is equal to $\lambda$. This means that the lowest drug concentration that guarantees bacterial eradication, $c_1$, is

determined by $\phi\,(c_1) = \lambda$. In our plate assay (**Figure 1B**), $c_1$ corresponds to the lowest antibiotic concentration at which no cells form colonies, that is, MIC; thus $c_1$ = MIC. Alternatively put, at the MIC, $\phi = \lambda$; see the main text for details.

Above the MIC ($\phi > \lambda$), all populations eventually undergo extinction (**Equation A6**), but the extinction time cannot be predetermined. **Equation A5** predicts that as time goes by, the probability of population extinction converges to $P_0(t) \approx 1 + \left(\frac{\lambda}{\phi} - 1\right) \cdot e^{(\lambda-\phi)\cdot t}$. In our experiments, we have counted the number of live colonies, i.e., colonies that have not gone extinct yet (i.e., $n \neq 0$). The probability of finding live colonies at time $t$, $P_{n\neq0}(t)$, which is equal to $1 - P_0(t)$, is

$$P_{n\neq0}(t) = 1 - P_0(t) \approx \left(1 - \frac{\lambda}{\phi}\right) \cdot e^{(\lambda-\phi)\cdot t}. \tag{A7}$$

Thus, at later times, the probability of finding live colonies can be approximated by an exponential decay.

Using **Equation A3**, moment dynamics can be derived (**Novozhilov et al., 2006**; **Pavel Krapivsky and Ben-Naim, 2010**; **Kendall, 1948**). The first moment, $\sum\limits_{n=0}^{\infty} nP$, is equal to the mean of $n$, $\langle n \rangle$, and the equation for the first moment is

$$\langle \dot{n} \rangle = \lambda \langle n \rangle - \phi \langle n \rangle. \tag{A8}$$

In our experiments, the initial condition, $n(0)$, is the same across the populations. For example, for all our microscope experiments, $n(0) = 1$.

Its solution is

$$\langle n(t) \rangle = n(0) \cdot e^{(\lambda-\phi)\cdot t}. \tag{A9}$$

These equation and solution are the same as the deterministic equation and solution we discussed above (**Equation A1 and A2**); therefore, the mean of the stochastic variable $n$ obeys the deterministic model.

Next, we consider the mean number of dead cells $\langle n_D \rangle$. Because live cells die at the rate of $\phi$, we have

$$\langle \dot{n}_D \rangle = \phi \langle n \rangle. \tag{A10}$$

Solving **Equation A10** together with **Equation A9**, we have

$$\langle n_D(t) \rangle = \frac{\phi \cdot \langle n(0) \rangle}{\lambda - \phi} \cdot \left[ e^{(\lambda-\phi)\cdot t} - 1 \right]. \tag{A11}$$

or alternatively,

$$\langle n_D(t) \rangle = \frac{\phi}{\lambda - \phi} \cdot \left[ \langle n(t) \rangle - n(0) \right]. \tag{A12}$$

We can also write an equation for the second moment, $\sum_{n=0}^{\infty} n^2 P_n$,

$$\frac{d\langle n^2 \rangle}{dt} = 2(\lambda - \phi)\langle n^2 \rangle + (\lambda + \phi)\langle n \rangle. \tag{A13}$$

The solution is

$$\langle n^2 \rangle = n(0)\frac{\lambda + \phi}{\lambda - \phi}e^{(\lambda-\phi)t}\left[e^{(\lambda-\phi)t} - 1\right] + n(0)^2 e^{2(\lambda-\phi)t}. \tag{A14}$$

Using **Equation A9 and S14**, we can calculate the variance $Var[n] = \langle n^2 \rangle - \langle n \rangle^2$,

$$Var[n] = n(0)\frac{\lambda + \phi}{\lambda - \phi} \cdot e^{(\lambda - \phi) \cdot t}\left[e^{(\lambda - \phi) \cdot t} - 1\right]. \tag{A15}$$

Next, we quantified population fluctuations by calculating the coefficient of variation (CV), that is, standard deviation relative to the mean. The standard deviation is equal to $\sqrt{Var[n]}$ and can be calculated using **Equation A15**. The mean is given by **Equation A9**. Thus, we have

$$CV = \frac{\sqrt{Var[n]}}{\langle n \rangle} \approx \sqrt{\frac{\lambda + \phi}{\lambda - \phi}} \cdot n(0)^{-1/2}. \tag{A16}$$

**Equation A16** holds for a sub-MIC region ($\lambda > \phi$) and considers the long-time limit. It indicates that the magnitude of fluctuations in population size depends on inoculum size ($n(0)$), and the rates of growth and death.

## Appendix 2

DOI: https://doi.org/10.7554/eLife.32976.026

To rule out heritable resistance as an explanation for heterogeneous colony formation (*Figure 1B*), we performed two rounds of plating at sub-MIC drug concentrations (*Figure 1C* and *Figure 1—figure supplement 3*). We picked a few colonies from the agar plate exhibiting a plating efficiency of ~0.5 (e.g., near 0.6 × MIC) and plated them immediately on fresh agar plates containing various concentrations of the same drug. The plating efficiency was similar or marginally lower on the second plating, possibly because exposure to the antibiotics on the first plate adversely affected the cells and rendered them more susceptible to the antibiotics on the second plating. This observation rules out heritable resistance as an explanation for heterogeneous colony formation at sub-MIC drug concentrations. Of course, if we perform multiple rounds of plating, we might eventually select for antibiotic-resistant mutants. However, for each round, we collect ~$10^5$ cells from a previous plate and plate ~100 cells on a fresh plate. This procedure creates a bottleneck effect, which reduces genetic variation and effectively weakens selection pressure for antibiotic resistance. Therefore, we expect that the evolution of antibiotic resistance is slow.

# Appendix 3

DOI: https://doi.org/10.7554/eLife.32976.027

We characterized the fluctuations in inoculum size by determining standard deviation and percent variation; see *Appendix 3—table 1*. The variation was ~10% or less. In *Figure 3D*, which shows the extinction probability at various (average) inoculum size, the probability changes only gradually for two-fold increment in inoculum size (which is equivalent to a 100% increase). Therefore, we do not believe that these fluctuations in inoculum size have a significant effect on the results.

**Appendix 3—table 1.** Percent variation of inoculum size in a microtiter plate.

| Average inoculum size | Standard deviation | Percent variation |
|---|---|---|
| 11.4 | 1.49 | 13% |
| 58.4 | 3.7 | 6.3% |
| 135.8 | 7.8 | 7.8% |

DOI: https://doi.org/10.7554/eLife.32976.028

