## [Decision Letter]

Thank you for submitting your article "Antibiotic-induced population fluctuations and stochastic clearance of bacteria" for consideration by *eLife*. Your article has been reviewed by three peer reviewers, and the evaluation has been overseen by a Reviewing Editor and Arup Chakraborty as the Senior Editor. The following individual involved in review of your submission has agreed to reveal his identity: Bartek Waclaw (Reviewer #3).

The reviewers have discussed the reviews with one another and the Reviewing Editor has drafted this decision to help you prepare a revised submission.

The reviewers found the topic very interesting, however they raise major concerns, both technical and in terms of the discussion of the results, that need to be addressed in great detail if publication is warranted. Please provide a detailed response to all points, the technical points of all reviewers, comparison between the model and experiments, discussion of previous work and the points about the impact of these findings, the parameter regimes where they occur and their link to real environments. While to the best of our knowledge the results presented here have never been reported in exactly this form, the lack of references to past work gives a false impression that no stochastic models of antibiotic resistance have been studied, which is not true (e.g., work from Hermsen and Hwa, the Dunlop group and others, additionally to the ones named by the reviewers (see point 5 below) build on similar ideas). This should be discussed in more detail, especially work that does try to make connections between theory and experiment. In view of these previous papers, the present result is maybe not extremely unexpected and should be presented more as an extension and careful regime exploration than a completely new idea. We therefore urge the authors to present the novelty of these results in a more balanced way.

Major points to be addressed:

1) All reviewers feel that the relevance of the conclusions to real infections is overstated. First, there is a relatively narrow window in which stochastic clearance occurs and it may be challenging to hit this precise window outside of a well-controlled lab experiment. Real infections are not treated when the number of cells is 100 to 10^4 but rather 10^8 or more since only an infection becomes evident. In such large populations different stochastic effects (de novo mutations that confer resistance to antibiotics, phenotypic adaptation) are likely to be more important than demographic fluctuations. Second, antibiotics only help to clear infections in vivo, and the most work is probably done by the immune system. Extrapolating from in vitro experiments where no immune response is present can be misleading. It would be interesting to discuss the possible role of the immune system in bacterial clearance within an actual patient. The authors should discuss these points in detail and tone down their statement.

2) Although the authors present data in Figure 1 showing this is not the case for the two rounds of exposure they use here, please discuss the effect of repeated sub-MIC antibiotic exposure and how this will ultimately impact the evolution of antibiotic resistance in the cells that do survive.

3) The manuscript would be improved by a more thorough and explicit comparison between the data and the model predictions:

- In Figure 1 and Figure 1—figure supplement 2, trends in the data have been highlighted by gray arrows. While we agree that the data show these trends, such "guides for the eye" can be misleading. Please graph linear regressions of the data (on the whole data set for Figure 1 and Figure 1—figure supplement 2, and for concentrations below a certain threshold for Figure 1). It would be useful to indicate the slope and the correlation coefficients obtained.

- Figure 3 and subsection “A simple, stochastic model of the population dynamics accounts for stochastic clearance of bacterial populations”, last paragraph: The data follow the predicted exponential decay at long enough times, but it would be interesting to push the analysis further:

- Does the slope of the data in this regime agree with the prediction from Equation S7?

- Can the delay before the onset of this decay be understood by the full expression of 1-*P*_0_, or are there extra complications at early times?

- Subsection “A population with large inoculum size is subject to stochastic clearance at sub-MIC drug concentrations”, last two paragraphs: Here the fluctuations in inoculum size could play a role in the stochasticity of the outcome, especially for small inoculum sizes. This point should be briefly discussed.

- Figure 4, Figure 4—figure supplement 2-4 and subsection “Alteration of the extinction probability to facilitate bacterial eradication at sub-MIC drug concentrations”, last three paragraphs: The data look quite deterministic, while the phenomenon is stochastic, and while Figure 1 (which corresponds to the same data as the white columns in Figure 4) included error bars corresponding to a standard deviation over several experimental replicates.

If only one replicate was considered for some experiments, this should be stated. If the mean over different replicates was plotted, it should be mentioned, and error bars should be added, similarly to Figure 1.

- For example, could the slope of the dashed line in Figure 3 be predicted from the model? Or Figure 3 and Figure 4 – do they quantitatively agree with model predictions? This should be fairly easy to do given the existing data on growth and death rates.

4) In some cases, it would be nice to show more data, especially if it already exists:

- "CV decreased with increasing inoculum size, supporting the expectation above": It would be nice to show the corresponding data.

- Video: it would be helpful to include other videos for comparison purposes, e.g. one without antibiotic and one with a bacteriostatic antibiotic (sub-MIC).

5) As noted in the summary, please reference and discuss previous work more exhaustively. The reviewers feel the novelty of the findings with respect to existing models is exaggerated. It is not true that current models of population dynamics of bacteria exposed to antibiotics are almost exclusively deterministic and that this work is the first to account for stochastic death and replication. A similar approach was used as early as in the 60's, see Nissen-Meyer, 1966. See also a more recent paper that combines stochastic modelling and experiments, Ferrante et al., 2005. in vitro

---

## [Author Response]

Major points to be addressed:1) All reviewers feel that the relevance of the conclusions to real infections is overstated. First, there is a relatively narrow window in which stochastic clearance occurs and it may be challenging to hit this precise window outside of a well-controlled lab experiment. Real infections are not treated when the number of cells is 100 to 10^4 but rather 10^8 or more since only an infection becomes evident. In such large populations different stochastic effects (de novo mutations that confer resistance to antibiotics, phenotypic adaptation) are likely to be more important than demographic fluctuations. Second, antibiotics only help to clear infections in vivo, and the most work is probably done by the immune system. Extrapolating from in vitro experiments where no immune response is present can be misleading. It would be interesting to discuss the possible role of the immune system in bacterial clearance within an actual patient. The authors should discuss these points in detail and tone down their statement.

We extensively modified the first three paragraphs in the Discussion to incorporate this comment. For example, we a) deleted the statement "…the present study is the first to show…"; b) acknowledged that a mature infection is a typical target of antibiotic treatment and contains many cells; c) cited prior work discussing stochasticity; and d) acknowledged the role of host immune systems.

2) Although the authors present data in Figure 1 showing this is not the case for the two rounds of exposure they use here, please discuss the effect of repeated sub-MIC antibiotic exposure and how this will ultimately impact the evolution of antibiotic resistance in the cells that do survive.We discussed this point in Appendix 2. Briefly, although multiple rounds of plating will ultimately lead to selection of antibiotic-resistant mutants, our plating procedure creates a bottleneck effect, which reduces genetic variation and effectively weakens selection pressure for antibiotic resistance.3) The manuscript would be improved by a more thorough and explicit comparison between the data and the model predictions:- In Figure 1 and Figure 1—figure supplement 2, trends in the data have been highlighted by gray arrows. While we agree that the data show these trends, such "guides for the eye" can be misleading. Please graph linear regressions of the data (on the whole data set for Figure 1 and Figure 1—figure supplement 2, and for concentrations below a certain threshold for Figure 1). It would be useful to indicate the slope and the correlation coefficients obtained.

Thank you for the suggestion. Linear regression analyses were performed, and results were plotted as a grey line in Figure 1 and Figure 1—figure supplement 2. The slope, intercept, and R-squared values were provided in the captions.

- Figure 3 and subsection “A simple, stochastic model of the population dynamics accounts for stochastic clearance of bacterial populations”, last paragraph: The data follow the predicted exponential decay at long enough times, but it would be interesting to push the analysis further:- Does the slope of the data in this regime agree with the prediction from Equation S7?

This is an interesting question. The number of live populations decreases exponentially when exposed to drug concentrations above MICs (Figure 3). Quantitative prediction of the exponential decay requires the determination of growth and death rates. But, we were not able to determine these rates for cells exposed to drug concentrations above the MIC, because cells died too quickly; we tried hard but were not able to monitor enough division/death events to reliably determine these rates. We plan to pursue this question in the future by developing high-throughput methods.

- Can the delay before the onset of this decay be understood by the full expression of 1-P_0_, or are there extra complications at early times?

In our experiments, we culture cells in drug-free liquid medium and then plate them on LB solid medium containing an antibiotic. The plating defines time zero. Upon exposure to an antibiotic, cells change their morphology. This change proceeds for some time before cell death takes place, indicating a delay in antibiotic killing. This delay is expected to result from influx/uptake of antibiotic molecules, binding of antibiotic molecules to their targets, etc. This delay leads to the initial plateau in the number of live populations (Figure 3), complicating the analysis.

- Subsection “A population with large inoculum size is subject to stochastic clearance at sub-MIC drug concentrations”, last two paragraphs: Here the fluctuations in inoculum size could play a role in the stochasticity of the outcome, especially for small inoculum sizes. This point should be briefly discussed.

Thank you for pointing this out. We characterized the fluctuations in inoculum size by determining standard deviation and percent variation; see the table below. The variation was ~10% or less. In Figure 3, which shows the extinction probability at various (average) inoculum size, the probability changes only gradually for two-fold increment in inoculum size (which is equivalent to 100% increases). Therefore, we do not believe that these fluctuations in inoculum size have a significant effect on the results. We added this discussion in Appendix 3 and added a pointer to this discussion in Figure 3 legend.

Average inoculum sizeStandard deviationPercent variation11.41.4913%58.43.76.3%135.87.87.8%

- Figure 4, Figure 4—figure supplement 2-4 and subsection “Alteration of the extinction probability to facilitate bacterial eradication at sub-MIC drug concentrations”, last three paragraphs: The data look quite deterministic, while the phenomenon is stochastic, and while Figure 1 (which corresponds to the same data as the white columns in Figure 4) included error bars corresponding to a standard deviation over several experimental replicates.If only one replicate was considered for some experiments, this should be stated. If the mean over different replicates was plotted, it should be mentioned, and error bars should be added, similarly to Figure 1.

We added errors bars in the figures and the statement “We performed two biological repeats and plotted the mean. The error bars represent one standard deviation from the repeats.” in the legends.

- For example, could the slope of the dashed line in Figure 3 be predicted from the model?

Yes, we agree that such a comparison would be useful. However, we were not able to make the comparison due to experimental limitations. The limitations were discussed in A4 above.

Or Figure 3 and Figure 4 – do they quantitatively agree with model predictions? This should be fairly easy to do given the existing data on growth and death rates.

In the main text, we characterized populations starting with single cells. We determined extinction probability, *P_0_,*using experiments and the model, which showed good agreement (Figure 2). Here, we compare the model predictions with experimental data plotted in Figure 3*P_0_*with higher inoculum size) and Figure 4*P_0_*with growth reduction).

First, we used the data shown in Figure 3 and plotted *P_0_*at various inoculum size for ~0.9× MIC of cefsulodin as red columns in the figure (on the right). We then plotted the model prediction as the black line, which shows a similar trend but somewhat underestimates *P_0_*. With respect to Figure 4, we can infer how well the model captures experimental data directly from the figure. We designed our experiments such that the growth rate is reduced by a half. Therefore, the model predicts a two-fold increase in extinction probability, meaning that the colored columns should be twice higher than the open columns. Figure 4 shows that the former is higher, but not always by two-fold. Therefore, the model predictions and experimental data show similar trends but do not exactly match.

We note that we used the Markov model to capture the most fundamental processes, and its prediction agreed well with the experimental data obtained with simple conditions (e.g., populations starting with single cells, Figure 2). However, the model does not include other details of antibiotic action (as discussed in the subsection “Alteration of the extinction probability to facilitate bacterial eradication at sub-MIC drug concentrations”), which can play a role in more complicated conditions, e.g., populations starting with many cells (Figure 3) or changing the growth rate (Figure 4). We believe this is why the model predictions and experimental data show similar trends but do not exactly agree. We plan to construct more realistic models in the future, as soon as this manuscript is published.

4) In some cases, it would be nice to show more data, especially if it already exists:- "CV decreased with increasing inoculum size, supporting the expectation above": It would be nice to show the corresponding data.We apologize that the description was not clear. In this sentence, we were describing the model prediction; we clarified this by directly citing the equation number. Unfortunately, we do not have the experimental data because the experiments were performed using 96-chamber microtiter plates and the number of cells in each chamber was not determined.- Video: it would be helpful to include other videos for comparison purposes, e.g. one without antibiotic and one with a bacteriostatic antibiotic (sub-MIC).Yes. We included these videos.5) As noted in the summary, please reference and discuss previous work more exhaustively. The reviewers feel the novelty of the findings with respect to existing models is exaggerated. It is not true that current models of population dynamics of bacteria exposed to antibiotics are almost exclusively deterministic and that this work is the first to account for stochastic death and replication. A similar approach was used as early as in the 60's, see Nissen-Meyer, 1966. See also a more recent paper that combines stochastic modelling and experiments, Ferrante et al., 2005.

As discussed in A1, we toned down our statement of novelty and cited previous studies including the ones recommended here.